# Relation of Dietary n-3 and n-6 Fatty Acid Intakes to Metabolic Syndrome in Middle-Aged People Depending on the Level of HbA1c: A Review of National Health and Nutrition Survey Data from 2014 to 2016

**DOI:** 10.3390/medicina58081017

**Published:** 2022-07-29

**Authors:** Seo-Woo Park, Do-Yeong Kim, Gyeong-Tae Bak, Dae-Sung Hyun, Sung-Kyung Kim

**Affiliations:** 1Department of Preventive Medicine, Wonju College of Medicine, Yonsei University, Wonju 26426, Korea; paw111111111111@gmail.com (S.-W.P.); ocean1555@naver.com (D.-Y.K.); dmdkdwbrma@yonsei.ac.kr (G.-T.B.); 2Institute of Occupational and Environmental Medicine, Wonju College of Medicine, Yonsei University, Wonju 26426, Korea; 3Department of Occupational and Environmental Medicine, Wonju College of Medicine, Yonsei University, Wonju 26426, Korea

**Keywords:** metabolic syndrome, insulin resistance, hemoglobin A1c, nutrition, omega-3 fatty acid, omega-6 fatty acid, ratio

## Abstract

*Background and Objectives:* The relation of dietary n-6 fatty acid to metabolic syndrome has not been examined and clearly defined. To improve health in the general population, this study was to investigate the role of n-3 and n-6 fatty acids in the reduction in metabolic syndrome and to observe changes in the effects of these fatty acids depending on the level of insulin resistance. *Materials and Methods:* This cross-sectional study utilized national health and nutrition survey data from 2014 to 2016. From the data, a relation of n-3 and n-6 fatty acid intakes to metabolic syndrome and Hemoglobin A1c (HbA1c)’s role in the relation was evaluated and analyzed for 4852 patients between 40 and 64 years old. Intake frequency of 112 nutrition and daily consumption amounts were identified, and intakes of n-3 and n-4 fatty acids were calculated from this data. Metabolic syndrome was determined for each participant using diagnostic standards for the Asian population published by the National Cholesterol Education Program. *Results:* Among the total 4852 subjects, 1583 (32.6%) had metabolic syndrome; 736 of 1875 (39.3%) males and 847 of 2977 (28.5%) females had the syndrome. In males, when their HbA1c was low (<5.4%), intakes of both n-3 and n-6 fatty acids were related to a 43–63% decreased prevalence of metabolic syndrome with significance, and a similar negative tendency was also observed in females. On the contrary, for both males and females, no statistically significant correlation was present when HbA1c was high. *Conclusion*: It was considered that consistent and regular dietary intakes of n-3 and n-6 fatty acids may contribute greatly to prevent or treat metabolic syndrome in healthy males with normal insulin sensitivity, but the effect of their dietary intakes was found to be limited in a group with strong insulin resistance. The conclusion of this study presents a valuable reference and knowledge to provide nutritional education to the general population.

## 1. Introduction

### 1.1. Need for Study

Fatty acids (FAs) are further subdivided into saturated fatty acids (SFAs), monounsaturated fatty acids (MUFAs), and polyunsaturated fatty acids (PUFAs) such as omega-3 and omega-6 (n-3 and n-6 fatty acids) depending on their compositions. The n-6 FA is abundantly found in food of animal origin, cooking oil, and various vegetable seeds, while the n-3 FA is largely concentrated in blue-backed fish, nuts, and green vegetables [1]. As the ratio of beef, pork, and poultry consumption has been rising in recent times, the n-6/n-3 ratio has also risen from 1:1 to 10:1 [2].

There have been many studies that examined and confirmed the correlation of n-3 and n-6 FAs to various chronic illnesses such as hypertension and obesity [3,4,5,6,7]. Both n-3 and n-6 FAs provide protective effects for chronic inflammatory diseases such as obesity, diabetes, and metabolic syndrome [8,9]. In addition, it has been reported that these FAs have a correlation to parameters that are related to glucose metabolism such as insulin and HbA1c in metabolic syndrome [10]. Hence, with these results in mind, there may be a relation between intake of n-6 FA and metabolic syndrome, and dietary intake of n-6 FA may provide some positive effect in the prevention and reduction in metabolic syndrome through lifestyle modification.

Metabolic syndrome is a complex chronic disease of adults that is found to have simultaneous metabolic abnormalities in three or more parameters including high TG, low high-density lipoprotein (HDL), hypertension, glucose metabolism abnormality, and abdominal obesity, and it is a well-known risk factor for cardiovascular diseases, cerebrovascular diseases, and chronic renal disease [11,12,13].

Presently, the known risk factors of metabolic syndrome are abdominal obesity, insulin resistance, excessive calorie intake, diminished physical activities, low education level, manual work, and nonstandard work; while psychological effects such as depression also have been found to have some adverse contributions to the syndrome [14,15,16,17,18]. In East Asia, middle-aged Japanese men with metabolic syndrome that did not require treatment showed an increase in all-cause mortality and CVD compared to a younger group [19]. Additionally, the prevalence of metabolic syndrome is higher in the middle-aged Korean population than in the younger population, particularly in the female group [20,21,22,23]. Insulin provides an inhibitory effect to the catabolism of fatty acids in fat tissues in order to store energy from food consumption. It is known that resistance to insulin, which leads to lowered effects on glucose and lipid metabolism, affects the prevalence of metabolic syndrome considerably [24,25,26]. Based on these findings, it has been reported that the effect of dietary n-3 and n-6 FA consumption on the syndrome can vary greatly among individuals, depending on their different levels of dysfunctional glucose metabolism [27].

Homeostatic model assessment–insulin resistance (HOMA-IR) and the quantitative insulin sensitivity check index (QUICKI) are frequently used factors that reflect insulin resistance, while HbA1c, which reflects overall glucose control of the last two to three months, is also known to have an indicatory value for insulin resistance [28].

### 1.2. Recent Study Trends

Regarding the activity of delta-6 and delta-5 desaturases, both n-6 and n-3 FAs increase the number of insulin receptors, enhance the affinity of insulin for the receptor, and decrease insulin resistance [29,30]. PUFA-derived anti-inflammatory lipid mediators such as lipoxin resolved modulating inflammation in various experimental models [31]. In particular, an increase in n-3 FA intake promoted the production of eicosanoids, which are involved in platelet aggregation, vasoconstriction, and inflammatory responses by influencing the FA composition on cell membranes [32]. Alpha-linoleic acid (ALA) and EPA, among n-3 FAs, were shown to correlate with lowering metabolic syndrome [33,34], and a cohort study reported that n-6 FA intake also had some preventative effects on metabolic syndrome [35]. Conversely, there was a cohort study that noted no correlation between n-3 FA intake and metabolic syndrome [36]. In addition, a different study noted the varied effects of the FA, in that it only reduced metabolic syndrome in males without showing any reduction in females [37]. Another meta-analysis resulted in only DHA among n-3 FA correlates with metabolic syndrome, while n-6 FA had no effect in reducing the syndrome [38]. However, a stratified analysis related to gender and insulin resistance was not performed in this previous study, and it was found that n-3 FA did not reduce the risk of metabolic syndrome in Asian races. Therefore to determine the relationship between metabolic syndrome and dietary intake of n-3 and n-6 FA, analysis under more diverse conditions in Asian races is necessary.

### 1.3. Study Purpose

Until now, previous studies have shown diverse and irregular relations of n-3 FA to metabolic syndrome. In addition, the level of insulin resistance’s role in the relationship between dietary intake of n-3 and n-6 FA and metabolic syndrome has been rarely investigated.

Thus, based on the previous literature, this study hypothesized that intake of the FAs would have a reducing effect on metabolic syndrome, and that this effect would vary according to each middle-aged individual’s insulin resistance. To examine the hypothesis, the current study reviewed national health and nutrition survey data that reflected the general middle-aged population. In addition, although HOMA-IR is a typical index for insulin resistance, since blood glucose levels were absent from the data, this study instead utilized HbA1c as the parameter to examine any variant relation to metabolic syndrome.

## 2. Materials and Methods

### 2.1. Study Design

This study was cross-sectional research of both males and females between the ages of 40 and 64 years to find a relation of their dietary n-3 and n-6 FA intakes to metabolic syndrome, depending on each middle-aged individual’s varying levels of insulin resistance. Because it was found that the effects of complex chronic diseases such as metabolic syndrome on health were greater in the young and elderly, we operationally defined the population 40 to 64 years old as middle-aged and in this age group [39,40].

### 2.2. Study Participants and Data Collection

The Korea National Health and Nutrition Examination Survey (KNHANES) is an ongoing cross-sectional survey of Korean civilians. Each year, 10,000 to 12,000 individuals from 4600 households are selected from the panel to represent the Korean population using multistage clustering and stratified random sampling methods. Selected households participate in a survey, and basic health and nutrition data are obtained through interviews and medical examinations. Among the participants (*n* = 23,080) of KNHANES between 2014 and 2016, those aged 40 to 64 years old (*n* = 8355) were chosen for the analysis. Among them, any individual whose one or more dependent variables such as blood pressure, fasting glucose level, waist circumference, TG, or HDL, which are used to determine the presence of metabolic syndrome (MetS) was missing were excluded (*n* = 7579), and those (*n* = 5669) without information on dietary consumption frequency were also eliminated from the study. In addition, anyone (*n* = 5651), whose surveys did not contain a record HbA1c or body mass index (BMI) was also removed, and those (*n* = 4912) with a history of myocardial and cerebral infarct and cancer and those (*n* = 4879) who were under treatment for cardiovascular diseases were also excluded. Females who were pregnant or breastfeeding at the time of the survey were removed (*n* = 4865). Finally, participants with a daily calorie intake of 500 to 6000 kcal were included in the study (*n* = 4852) [41]. The final samples contained 1875 males and 2977 females, and their national health and nutrition survey data were reviewed and analyzed (Figure 1). The Institutional Review Board (IRB) of Yonsei University Wonju Severance Hospital approved this study (IRB No. CR320365).

### 2.3. Study Tools

#### 2.3.1. Evaluating Variables

(1)Variables Related to Metabolic Syndrome

Blood-related parameters such as fasting blood glucose, HbA1c, cholesterols, TG, systolic and diastolic blood pressures, and waist circumference were directly taken from the survey, and the mean blood pressures of the second and third measurements were used for the analysis. Measurements of HbA1c were divided into tertile groups from low to high, and the boundary values between each group were 5.4 and 5.7.

(2)Variables for Dietary Intake

In this study, individual energy and nutrient intake data obtained through the 24-h recall method during the nutrition section of the National Health and Nutrition Examination Survey were used. The 24-h recall survey was conducted through individual interviews, with the nutrition survey team visiting households about one week after the completion of the checkup survey. Energy intake was presented in kcal units consumed per day. The amount of each dietary intake was collected from dietary food consumption frequency data of the national health and nutrition survey. This data investigated and recorded intake frequencies of 112 varieties of nutrition by using a validated semiquantitative table for food intake frequency and each nutritional intake amount per meal. Among them, daily calorie consumption and amounts of protein, carbohydrates, SFA, n-3 and n-6 FAs, MUFA, cholesterol, and dietary fiber intake were utilized for the study’s analysis. Using the amount of n-3 and n-4 FAs, the n-6/n-3 FA ratio was calculated, and the amount of these two FAs and the ratio were divided into tertile groups to be used as independent variables.

(3)Variables for Lifestyles and Demography

Based on the data of the national health and nutrition survey, the participants who selected ‘smoking daily’ or ‘social smoker’ were identified as current smokers. Those who checked ‘not applicable’ were considered as nonsmokers, while those who chose ‘had smoked, but discontinued’ were classified as ex-smokers. For frequency of alcohol consumption, those who answered ‘had not drunk for the past one year’, ‘less than one per month’, ‘once per month’, or ‘not applicable’ were categorized as ‘0–1/month’. Those with alcohol consumption of two to four times per month were identified as ‘2–4/month’, while the participants who selected ‘two to three times per week’ or ‘four times or more per week’ were classified as ‘2 or more/week’. Educational levels were divided in accordance with final graduations. Anyone who identified their final graduations as ‘graduated from elementary school’, ‘graduated from middle school’, or ‘graduated from high school’ were classified as ‘below high school’, while those, who selected ‘graduated from college or above’ were identified as ‘university’. The level of physical activities was evaluated after asking the subjects questions using a calculation method suggested by the national health and nutrition survey (number of days they participated in walking exercise in a week × 3.3 + number of days they participated in moderately demanding exercise × 4 + number of days they participated in highly demanding exercise × 8 = physical activity level). The status of employment was divided into ‘employed’, ‘unemployed’, and ‘housework’. For the level of income, those who selected ‘poor’ were categorized as ‘poor’, and those who answered ‘above poor’, ‘above average’, or ‘high’ were all classified as ‘above poor’. For the occupation classification, the participants who selected ‘night working schedule’, ‘regular day/night shift working schedule’, ‘24-h shift working schedule’, or ‘irregular shift working schedule’, were considered night/shift workers; and those who answered ‘day working schedule’, ‘evening working schedule’, ‘split shift working schedule’, or ‘other working schedule’ were classified as non-night/shift workers.

#### 2.3.2. Evaluating Metabolic Syndrome

This study considered participants with metabolic syndrome who satisfied three or more of the following diagnostic standards published by the National Cholesterol Education Program (NCEP) in 2005 [32]. For male and female waist circumferences, the standards set by the Korean Society for the Study of Obesity were followed [22,42].

Waist circumference: male ≥ 90 cm, female ≥ 85 cm;TG: ≥150 mg/dL;HDL cholesterol: male ≥ 40 mg/dL, female ≥ 50 mg/dL;Blood pressure: systolic ≥ 130 mmHg or diastolic ≥ 85 mmHg;Fasting blood glucose: ≥100 mg/dL.

In addition, any participant who had been treated for diabetes, hypertension, or hyperlipidemia during the survey was automatically considered to have satisfied the fasting blood glucose, blood pressure, or HDL cholesterol/TG categories defined according to the respective diagnostic classifications mentioned above.

#### 2.3.3. Defining Confounding Variables

Age, smoking, alcohol consumption, occupational status, physical activity level, income, and daily intake amounts of calories, carbohydrates, protein, SFA, MUFA, cholesterols, and dietary fiber were defined as confounding variables, and they were used during a logistic regression analysis to ensure the validity of the outcomes.

#### 2.3.4. Statistical Analysis

The statistical significance was set *p* < 0.05; statistical analysis of the data was performed using SPSS Statistics for Windows, version 23.0 (IBM Corp., Armonk, NY, USA). The numbers, percentages, means, and standard deviations for the demographic characteristics and amounts of nutritional intakes were calculated using descriptive statistics. The presence of any correlation between nutritional intakes and demographic characteristics with metabolic syndrome was examined with a chi-squared test and an independent-sample *t*-test. To obtain the odds ratio (OR) and 95% confidence interval (CI) of the relation between each tertile group for n-3 FA, n-6 FA, and the n-6 FA/n-3 FA ratio and the prevalence of metabolic syndrome, binominal logistic regression analysis was utilized, and the analysis was adjusted using variables such as age, smoking, alcohol consumption, educational level, occupation, physical activity level, income, and daily intakes of calorie, carbohydrates, protein, SFA, MUFA, cholesterols, and dietary fiber. In analyzing the relation of each tertile group for n-3 FA, n-6 FA, and the n-6/n-3 ratio to the prevalence of metabolic syndrome in accordance with the different levels of HbA1c, binominal logistic regression was also utilized to calculate the odds ratio and 95% confidence interval, and they were deduced using a multiple regression analysis in accordance with confounding variables such as age (model 1), smoking, alcohol consumption, educational level, occupation, physical activity level, income (model 2), and daily intake of calorie, carbohydrates, protein, SFA, MUFA, cholesterols, and dietary fiber (model 3). Although all independent variables did not satisfy normality, the logistic regression analysis was not mentioned separately because it is an analysis that can be used even if the assumption of normality of the independent variable is not satisfied [43].

## 3. Results

Table 1 lists the relations of the demographic characteristics and dietary intake to the prevalence of metabolic syndrome in each male and female group. Among 4852 participants, there were 1875 males (38.64%) and 2977 females (61.36%). In males, all demographic characteristics except for physical activity level, night/shift work, and daily calorie consumption from dietary intake showed statistically significant differences. In females, all demographic characteristics except for smoking and night/shift work and all daily dietary intakes except for n-6 FA/n-3 FA ratio resulted in a statistically significant difference. In males, when age was compared between groups with and without metabolic syndrome, no difference was observed. On the contrary, for females, the participants with metabolic syndrome were about 4 years older than the subjects without metabolic syndrome, and this showed statistical significance (*p <* 0.001). The BMI and HbA1c levels were statistically higher in men and women with metabolic syndrome than in those without the syndrome (*p* < 0.001). For physical activity level, lower physical activity was found to have statistical significance only in females (*p <* 0.001), while night/shift work presented no statistical difference in both males and females. A significant difference in smoking was only presented in males (*p <* 0.001). Alcohol consumption was found have a statistical difference in both males and females (*p <* 0.001, *p* = 0.008). Although males with metabolic syndrome had a significantly higher consumption of alcohol, females showed a contradictory result: those with less frequent alcohol consumption were found to be more significantly affected by metabolic syndrome.

For educational level, a lower level was more significantly prevalent for the syndrome in both males and females (*p <* 0.001). For occupational activity, unemployment and housework resulted in a more statistically significant prevalence of the syndrome (*p <* 0.025, *p <* 0.001), while a lower income was found to be more significant in both males and females (*p* = 0.005, *p <* 0.001).

Analysis of dietary intake presented dissimilar outcomes between males and females. For males, only the total daily calorie intake had a statistically significant difference (*p <* 0.013), and those with metabolic syndrome consumed a considerably higher amount of calories. On the contrary, metabolic syndrome was more significantly prevalent in females, who had lower daily calorie, carbohydrate, and protein intakes (*p <* 0.001, *p <* 0.027, *p <* 0.001). In addition, for females with metabolic syndrome, fat consumption was significantly lower (*p <* 0.001), and each fatty acid (SFA, MUFA, PUFA, n-6 FA, n-3 FA, and cholesterols) was also consumed considerably less by females with metabolic syndrome (all *p <* 0.001).

Intakes of n-3 FA and n-6 FA and the ratio of n-6 FA intake/n-3 FA intake in each group of males and females were divided into a tertile group. Table 2 lists the comparisons of these intakes and ratios to the presence of metabolic syndrome and other variables, and their odds ratios were calculated. Additionally, Figure 2 illustrated associations between a tertile increase in PUFA intake and prevalence of metabolic syndrome. In males, boundary values of n-6 FA intake between groups 1 and 2 and between groups 2 and 3 were 6.4 g/day and 10.1 g/day, respectively; while boundary values of n-3 FA intake between groups 1 and 2 and between groups 2 and 3 were 0.8 g/day and 1.4 g/day, respectively. Boundary values of the ratio between groups 1 and 2 and between groups 2 and 3 were 6.9 and 7.8, respectively, in males. For females, boundary values of n-6 FA intake between groups 1 and 2 and between groups 2 and 3 were 5.6 g/day and 8.9 g/day, respectively; while boundary values of n-3 FA intake between groups 1 and 2 and between groups 2 and 3 were 0.8 g/day and 1.3 g/day, respectively. Boundary values of the ratio between groups 1 and 2 and between groups 2 and 3 were 6.6 and 7.5, respectively, in females.

The males who met the blood pressure criteria of metabolic syndrome showed 0.77 times (95% CI: 0.59–0.99) less intake of n-6 FA in the second tertile group, and this was found to be statistically significant. In addition, for the second tertile group, these participants, who were diagnosed with the syndrome and had increased waist circumference, showed 0.814 times (95% CI: 0.63–1.06) and 0.771 times (95% CI: 0.59–1.01) less intake of n-6 FA, respectively, but these presented no significant differences.

Intake of n-3 FA in males showed that the participants in the second tertile group with metabolic syndrome (OR: 0.70; 95% CI: 0.54–0.91), high blood pressure (OR: 0.71; 95% CI: 0.55–0.92), and high fasting blood glucose (OR: 0.73; 95% CI: 0.57–0.95) resulted in a significantly lowered intake. The ratio of n-6 FA/n-3 FA presented a lower value in the second tertile group when fasting blood glucose (OR: 0.86; 95% CI: 0.67–1.09) met the diagnostic criteria, but it was not found to have a statistical significance.

In females, n-6 FA intake showed that when HDL cholesterol within a second (OR: 0.78; 95% CI: 0.63–0.96) and third (OR: 0.71; 95% CI: 0.518–0.97) tertile group met the criteria, a statistically significant reduction was present. When metabolic syndrome and waist circumference were met within the second tertile group, they also decreased to 0.85 (95% CI: 0.67–1.067) and 0.84 (95% CI;:0.66–1.06) times, respectively, without showing any statistical significance.

Intake of n-3 FA was noted to be significantly less when waist circumference (OR: 0.79; 95% CI: 0.63–0.99), fasting blood glucose (OR: 0.78; 95% CI; 0.62–0.97), and HDL cholesterol (OR: 0.75; 95% CI: 0.61–0.92) were satisfied in the females’ second tertile group. For the third tertile group in females, the intake decreased significantly when only criteria for HDL cholesterol (OR: 0.67; 95% CI: 0.50–0.90) was met. Although no significant difference was present, when metabolic syndrome (OR: 0.85; 95% CI: 0.68–1.06) and TG (OR: 0.82; 95% CI: 0.66–1.02) were fulfilled, the second tertile group of females also showed a decrease in n-3 FA.

The ratio of n-6 FA/n-3 FA intake was significantly lowered in second tertile group of females when the criteria of metabolic syndrome (OR: 0.79; 95% CI: 0.64–0.97), blood pressure (OR: 0.74; 95% CI: 0.61–0.90), waist circumference (OR: 0.79; 95% CI: 0.64–0.98), and fasting blood glucose (OR: 0.81; 95% CI: 0.66–0.99) were satisfied. The ratio was also significantly lowered in the females’ third tertile group when the participants only fulfilled the criteria for blood pressure (OR: 0.72; 95% CI: 0.58–0.91). In addition, HDL cholesterol (OR: 0.85; 95% CI: 0.70–1.02) and TG (OR: 0.82; 95% CI: 0.67–1.00) in the second tertile group and fasting blood glucose (OR: 0.81; 95% CI: 0.64–1.01) in the third tertile group showed a lowered ratio, but these presented no statistical significance.

Table 3 lists the relation of metabolic syndrome to n-3 FA and n-6 FA depending on HbA1c, which was classified into a tertile group. Additionally, Figure 3 showed associations between a tertile increase in PUFA intake and prevalence of metabolic syndrome according to HbA1c level. In males and females, the boundary values of the tertile groups were 5.4% and 5.7%, respectively. Since the diagnostic criteria for diabetes state that less than 5.7% is normal, from 5.7 to 6.4% is prediabetes, and more than 6.4% is diabetes, the upper tertile groups were considered to be in a prediabetic or diabetic state. As shown in Table 2, for each group of males and females, the intake of each fatty acid was stratified with HbA1c and also categorized into a tertile group. They were compared to the presence of metabolic syndrome, and ORs were calculated.

In males, when the participants had n-6 FA intake of the second and third tertiles with both groups in the first tertile of HbA1c, the prevalence of metabolic syndrome depending on n-6 FA intake was found to be 0.57 (95% CI: 0.33–0.99) and 0.42 (95% CI: 0.18–0.96) times less, respectively, and they showed statistical significance. Although the prevalence of the syndrome was 0.94 (95% CI: 0.57–1.55) and 0.87 (95% CI: 0.57–1.34) times in the second tertile of n-6 FA with both the second and third tertiles of HbA1c, respectively, the comparison resulted in no significant differences.

When the n-3 FA intake of males was compared to metabolic syndrome, the results were similar to the comparison of n-6 FA to the syndrome. For the first tertile of HbA1c, both the second and third tertiles of n-3 FA were significantly reduced to 0.466 (95% CI: 0.269–0.805) and 0.367 (95% CI: 0.16–0.82) times, respectively. In addition, the prevalence of the syndrome was reduced to 0.67 (95% CI: 0.46–1.06) and 0.85 (95% CI: 0.45–1.62) times for third tertile of HbA1c within both the second and third tertiles of the FA intake, respectively, but these showed no statistical significance. The ratio of n-6 FA/n-3 FA in males resulted in ORs of 0.75 (95% CI: 0.47–1.20) and 0.79 (95% CI: 0.47–1.33) for the second tertile of HbA1c in both the second and third tertiles of the ratio, but these also did not result any significant differences.

In females, the first tertile of HbA1c in n-3 and n-6 FAs showed a tendency toward a lowered prevalence in metabolic syndrome, but no comparison had any statistical difference. The n-6 FA tertiles with the first tertile of HbA1c had lowered tendencies of 0.72 (95% CI: 0.43–1.18) and 0.65 (95% CI: 0.30–1.43) times in sequence, and n-3 FA tertiles with the first tertile of HbA1c also showed decreased OR tendencies of 0.83 (95% CI: 0.51–1.35) and 0.63 (95% CI: 0.30–1.32) times in sequence. The second and third tertiles of HbA1c again resulted in no statistical significance when compared, and the only negligible tendency was shown for n-6 intake in the second tertile of HbA1c and n-3 intake in the third tertile of HbA1c.

## 4. Discussion

This study investigated and compared the relation of dietary n-3 and n-6 FA intakes to metabolic syndrome, depending on varied insulin resistance, in order to provide valuable information on preventing and managing this adverse syndrome in the aspect of modifying a lifestyle. In females with metabolic syndrome, the intake of total calories, protein, fat, SFAs, MUFAs, PUFAs, n-6 FA, n-3 FA, cholesterol, and total fiber were significantly lower than that of females without metabolic syndrome. Only carbohydrate intake did not show a difference, which may have been a result of reflecting the dietary characteristics of the female metabolic syndrome group. Our findings were consistent with analyses that used 2011 KNHANES data, which found that the risk of metabolic syndrome in women was only higher when the proportion of carbohydrate intake was high and the proportion of fat intake was low [44].

The outcomes of the current study showed a lowered risk of metabolic syndrome in males with little insulin resistance and a large dietary intake of n-3 and n-6 FAs. Insulin resistance greatly affects the prevalence of the syndrome, and it has been reported that the effect of these FAs on metabolic syndrome varied depending on the different levels of insulin resistance [27]. One Cochrane study stated that n-3 FA intake lowered serum TG levels in patients with type 2 diabetes, but it may elevate LDL, and had no useful effect on glucose control [45]. In another larger study on diabetic patients, it was found that there were vague results regarding whether n-3 FA intake had any effect on LDL, and this study also noted that the intake had no effect of increasing insulin sensitivity in diabetic patients, while people without diabetes had the beneficial effect of raised insulin sensitivity due to intake of the FA [46,47,48,49]. These studies supported the outcomes of the current study, in which dietary intakes of n-3 and n-6 FAs in the participants with a high insulin resistance did not have any significant correlation with metabolic syndrome. Hence, we supposed that insulin resistance is more influential in the prevalence of the syndrome than dietary intake of these FAs.

A previous meta-analysis was conducted that had results similar to the results of the current study. Like the results of the meta-analysis, the results found in the male group of this study showed negative correlations between n-3 FA and metabolic syndrome, while intake of n-6 FA had no significant correlation to the syndrome [38]. However, according to a cross-sectional study that was similar to the design of the current study, n-6 FA presented a preventative effect in the prevalence of metabolic syndrome [33]. As dietary intake of n-6 FA increased, the prevalence was reduced, and a higher intake led to a lowering effect on the TG level. Likewise, there have been contradictory studies that reported opposite roles of n-6 FA in inflammation [50,51,52].

The current study showed that the second tertile of n-3 and n-6 FAs had a correlation with hypertension, and this finding was similar to a that of previous cross-sectional study [6]. The only difference was that, in the previous study, the third tertile also correlated with a reduction in hypertension prevalence. We considered that the reason the third tertile of this study did not have a correlation was that a large intake of n-3 and n-6 FAs was accompanied by a higher possibility of consuming a large amount of SFAs and this resulted in some effects on the metabolism of n-3 and n-6 FAs. Although there is not a definitive explanation of n-3 and n-6 FA intakes in reducing the risk of hypertension, it is believed that, since increased inflammatory reaction plays a significant role in the prevalence of hypertension, the known anti-inflammatory effect of n-3 FA may be able to reduce the risk of hypertension. In addition, n-3 FA enhances the vasodilation of endothelial cells and reduces the function of resistant vascular smooth muscle [53,54]. Finally, n-3 and n-6 FAs function as signal molecules to peroxisome proliferator-activated receptors (PPARs) that control various metabolic processes, including inflammatory reactions, oxidation stress, formation of lipocytes, and metabolism of lipids and glucose [55].

It has been suggested that an imbalanced diet with a high n-6/n-3 FA intake ratio is a very important risk factor for elevated blood lipid dysfunction, insulin resistance, and heart disease in metabolic syndrome [50,56]. In this study, a clinically significant association was found in metabolic syndrome with an n-6/n-3 FA ratio in the third tertile or higher for both men and women in the glycated hemoglobin category above the third tertile, respectively, while the lowest n-6/n-3 FA ratios were associated with an increased risk of metabolic syndrome for an OR of 1.59 (95% CI: 1.00–2.52) and an OR of 1.52 (95% CI: 1.03–2.23) compared to the first tertile category. In the case of women, the risk increased at a lower n-6/n-3 ratio than in men, so it may have a more important meaning in women in the high-level glycated hemoglobin group. However, in both men and women, in the glycated hemoglobin category of less than the third tertiary, n-6 was as effective as n-3 FA in preventing metabolic syndrome in the low group. This finding in the current study was also comparable to those of other previous studies [6,33,34,57]. According to a study by Cicero et al., in 2013, the effective intake of omega-3 was 2 to 6 g per day [58]. In this study, in the case of n-6 or n-3 fatty acids in men with normal blood sugar, intakes of 6.4 g or 0.9 g or more per day, respectively, was found to reduce the risk of metabolic syndrome. Although not statistically significant in women, the doses of n-6 or n-3 fatty acids at 5.6 g or 0.8 g or more per day, respectively, were found to reduce the risk of metabolic syndrome. Therefore, in middle-aged adults with normal blood sugar, the total daily dose for reducing the risk of metabolic syndrome was not considered to be significantly different from previous studies. In the hyperglycemic group, high-dose PUFA intake and the n-6/n-3 fatty acid ratio were also found to be important factors. However, high-dose PUFA intake may be detrimental to health, depending on the underlying disease, and a doctor’s consultation is required [58]. However, in this study, as the interval was set as the tertile, it is necessary to further subdivide the interval in future studies to calculate the risk for each intake dose.

Contrary to the findings for males, females showed no significant correlation of n-3 and n-6 FAs to metabolic syndrome. Many preceding studies did not divide their analysis according to gender, but one previous study that more specifically analyzed the correlation between n-3 FA and metabolic syndrome within each gender of the Korean population, discovered that n-3 FA intake in females did not play a statistically significant role in lowering the risk of the syndrome [37]. To explain this insignificant correlation, this study suggested that the level of n-3 or n-6 FAs in some females may not actually be sufficient enough to have a treatment value, even though their subjects had reported a substantial intake amount of the FAs. According to the outcomes from the current study, it is possible to interpret that those without metabolic syndrome are likely to take in more n-3 and n-6 FAs, since it was found that the participants with metabolic syndrome had consumed significantly lower amounts of daily FAs. In addition, when considering the finding that the participants with metabolic syndrome had a lower amount of daily calorie intake and lower FA consumption, it is possible to expect that this group had already been performing a strict diet control, as they understood many risks originating from metabolic syndrome. To re-evaluate the correlation of the dietary FA intake with metabolic syndrome in females, and to more specifically examine the correlations already found in males, the authors of this study consider it necessary to plan and implement further studies in the future.

There were several limitations to the current study. Due to its nature as a cross-sectional study, this study failed to prove causal correlations between n-3 and n-6 FA intakes and metabolic syndrome in accordance with varying levels of insulin resistance. In addition, since only 11.7% of males and 7.5% of females had a level of HbA1c of 6.5% or more and were classified as diabetics, the number of these diabetic patients was too small to form groups with or without diabetes for further detailed comparison (Appendix A). If a cohort study is designed in such a way to divide participants into diabetic and nondiabetic groups, this additional examination of the relation of n-3 and n-6 FA intakes to metabolic syndrome in accordance with the presence of the disease may also provide more information on their correlation. Additionally, for complex diseases that include various symptoms or diseases, such as metabolic syndrome, there is a possibility that bias may occur in each test or measurement step, and information on people taking dietary supplements such as vitamin E, vitamin D, and antioxidants may not have been included in the analysis, which may have served as a bias.

Because this study used data only from a national health and nutrition survey, those who were consuming supplementary n-3 FA could not be identified and excluded for a more precise analysis. However, the frequency of using the supplement is known to be minimal, and its effect on this study’s analysis was considered to be negligible. Although several previous studies found and noted that varied components of n-3 and n-6 FAs affect metabolic syndrome at different levels [33,36,38], the inability to investigate constituents of n-3 and n-6 FAs due to reviewing the reported dietary frequency surveys was another limitation of the current study. Finally, this study was not able to exclude participants with conditions of abnormal hemoglobin, blood disorder with anemia, or diseases in the kidney or liver, which may adversely influence the accuracy of HbA1c measurements. In addition, it should be taken into consideration in future research that the likelihood of insulin resistance was determined only by the HbA1c level, and there is a probability that non-insulin resistance states such as type 1 diabetes were also included in the study subjects with an abnormal HbA1c status. Lastly, many studies have suggested a relationship between elevated liver levels and metabolic syndrome, and liver enzyme or urine tests may be necessary to consider the association with diseases such as chronic kidney disease. In particular, the benefits of PUFA intake in the case of high liver levels are well known [59]. Treatment in the case of elevated liver enzyme levels and metabolic syndrome may be effective, and future prospective studies will expand our findings.

Nonetheless, the current study was still considered valuable, as it was the first study to examine the correlation between n-6 FA and metabolic syndrome in the Korean middle-aged population, and the study uncovered that intakes of n-3 and n-6 FAs in males with a high insulin sensitivity significantly lowered the prevalence of the syndrome. This study was also the first to suggest that, to the best of our knowledge, a high n-6/n-3 fatty acid ratio in the high-glycated hemoglobin group was associated with an increased prevalence of metabolic syndrome in both middle-aged men and women in Korea. In addition, it had the advantage of correcting the odds ratio by setting occupational factors such as shift work and employment status as confounding variables, which have recently been found to be related to metabolic syndrome. It is necessary to conduct more similar investigations in different ethnic or population groups to supplement the outcomes of this study, since its study subjects were limited to the Korean population.

## 5. Conclusions

Regular and consistent dietary intake of n-3 and n-6 FAs in healthy middle-aged males with an HbA1c level of 5.4% or less may provide a beneficial role in preventing or treating metabolic syndrome. On the contrary, this diet did not have any effect on the syndrome in groups with elevated insulin resistance. Thus, it was considered that a high degree of insulin resistance had a substantial influence on metabolic syndrome.

## Figures and Tables

**Figure 1 medicina-58-01017-f001:**
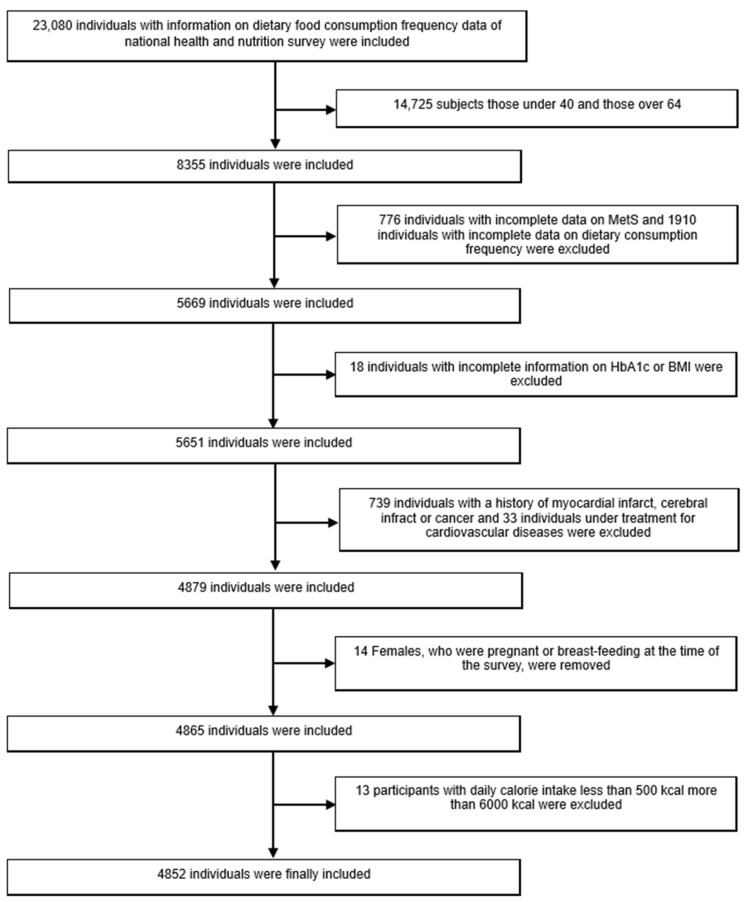
The population of the study.

**Figure 2 medicina-58-01017-f002:**
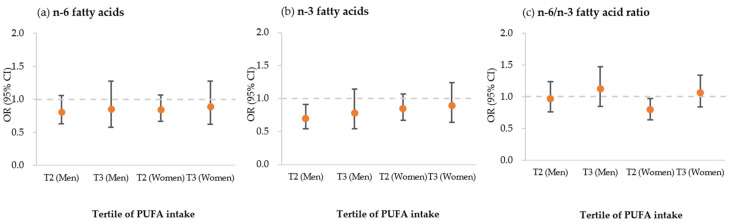
Associations between a tertile increase in PUFA intake and prevalence of metabolic syndrome for (**a**) n-6 fatty acid, (**b**) n-3 fatty acid, and (**c**) the n-6/n-3 ratio using the odds ratio (OR). ORs were adjusted for age, smoking status, alcohol use frequency, education level, employment status, lowest income quartile, physical activity, and daily intake (energy, carbohydrates, protein, saturated fatty acid, monounsaturated fatty acid, cholesterol, and dietary fiber). Error bars represent the 95% confidence interval for each point estimate. Tertile values for each PUFA intake are provided in Table 2.

**Figure 3 medicina-58-01017-f003:**
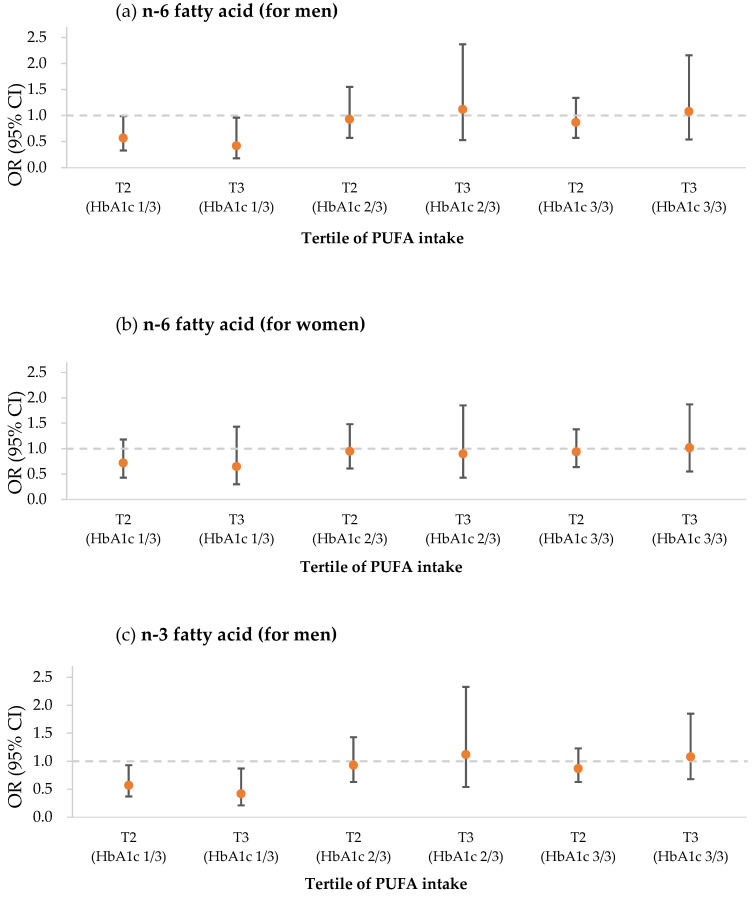
Associations between an tertile increase in PUFA intake and prevalence of metabolic syndrome according to HbA1c level in (**a**) n-6 fatty acid (for men), (**b**) n-6 fatty acid (for men), (**c**) n-3 fatty acid (for men), (**d**) n-3 fatty acid (for women), (**e**) the n-6/n-3 fatty acid ratio (for men), and (**f**) the n-6/n-3 fatty acid ratio (for women) using the odds ratio (OR). ORs were adjusted for age, smoking status, alcohol use frequency, education level, employment status, lowest income quartile, physical activity, and daily intake (energy, carbohydrates, protein, saturated fatty acids, monounsaturated fatty acids, cholesterol, and dietary fiber). Error bars represent the 95% confidence interval for each point estimate. Tertile values for each PUFA intake and HbA1c level are provided in Table 3.

**Table 1 medicina-58-01017-t001:** Characteristics and dietary intakes of participants by gender and metabolic syndrome status.

	Men (*n* = 1875)			Women (*n* = 2977)		
	Non-MetS	MetS	*p*-Value	Non-MetS	MetS	*p*-Value
Participants (*n*)	1139	736		2130	847	
Age	51.2 ± 7.2	52.9 ± 7.0	<0.001	50.6 ± 6.9	54.7 ± 6.6	<0.001
BMI	23.6 ± 2.6	26.2 ± 3.1	<0.001	22.7 ± 2.7	26.3 ± 3.5	<0.001
Physical activity index	24.2 ± 21.6	22.8 ± 21.8	0.154	22.1 ± 19.2	18.1 ± 14.8	<0.001
HbA1c	5.6 ± 0.6	6.14 ± 1.1	<0.001	5.5 ± 0.5	6.1 ± 1.0	<0.001
Smoking status (*n*, %)			<0.001			0.062
Current	414 (36.3%)	294 (39.9%)		81 (3.8%)	47 (5.5%)	
Non	249 (21.9%)	105 (14.3%)		1974 (92.7%)	777 (91.7%)	
Ex	476 (41.8%)	337 (45.8%)		75 (3.5%)	23 (2.7%)	
Drinking frequency (*n*, %)			<0.001			0.008
0~1/month	424 (37.2%)	212 (28.8%)		1448 (68%)	625 (73.8%)	
2–4/month	326 (28.6%)	184 (25.0%)		432 (20.3%)	141 (16.6%)	
More than 2/week	389 (34.2%)	340 (46.2%)		250 (11.7%)	81 (9.6%)	
Educational level (*n*, %)			<0.001			<0.001
Below high school	584 (51.3%)	450 (61.1%)		1422 (66.8%)	695 (82.1%)	
University	555 (48.7%)	286 (38.9%)		708 (33.2%)	152 (17.9%)	
Occupational status (*n*, %)			0.025			<0.001
Unemployed	107 (9.4%)	97 (13.2%)		825 (38.7%)	363 (42.9%)	
Employed	1017 (89.3%)	626 (85.1%)		1193 (56%)	406 (47.9%)	
Housework	15 (1.3%)	13 (1.8%)		112 (5.3%)	78 (9.2%)	
Night/shift work (*n*,%)			0.584			0.816
Yes	82 (7.2%)	58 (7.9%)		56 (2.6%)	21 (2.5%)	
No	1057 (92.8%)	678 (92.1%)		2074 (97.4%)	825 (97.5%)	
Lowest income quartile (*n*, %)	69 (6.1%)	70 (9.5%)	0.005	172 (8.1%)	139 (16.4%)	<0.001
Daily dietary intake						
Total energy (kcal)	2144.7 ± 690.9	2230.8 ± 757.7	0.013	1698.9 ± 591.7	1621.0 ± 570.8	0.001
Carbohydrate (g)	339.4 ± 98.7	339.3 ± 104.0	0.975	282.8 ± 94.6	278.2 ± 92.0	0.227
Protein (g)	66.67 ± 27.1	67.7 ± 30.3	0.471	57.8 ± 24.7	52.9 ± 22.8	<0.001
Fat (g)	38.6 ± 21.0	38.2 ± 22.1	0.67	33.7 ± 18.7	28.3 ± 17.1	<0.001
SFA (g)	11.3 ± 6.5	11.7 ± 6.6	0.515	9.4 ± 5.47	7.8 ± 5.0	<0.001
MUFA (g)	12.0± 7.0	11.9 ± 7.4	0.865	10.1 ± 6.0	8.4 ± 5.57	<0.001
PUFA (g)	10.2 ± 5.4	10.2 ± 5.9	0.904	9.5 ± 5.3	8.18 ± 4.8	<0.001
n-6 FA (g)	9.1 ± 4.8	9.1 ± 5.3	0.943	8.37 ± 4.7	7.2 ± 4.3	<0.001
n-3 FA (g)	1.25 ± 0.7	1.3 ± 0.8	0.812	1.2 ± 0.7	1.06 ± 0.6	<0.001
Cholesterol (mg)	241.5 ± 152.3	240.9 ± 160.8	0.938	224.2 ± 148.9	190.9 ± 144.9	<0.001
Total fiber (g)	20.7 ± 8.7	21.1 ± 9.6	0.307	21.1 ± 9.4	20.3 ± 8.7	0.048
n-6/n-3 FA ratio	7.5 ± 1.6	7.4 ± 1.5	0.543	7.1 ± 1.4	7.0 ± 1.5	0.056

BMI, body mass index; HbA1c, glycated hemoglobin; SFA, saturated fatty acid; MUFA, monounsaturated fatty acid; PUFA, polyunsaturated fatty acid.

**Table 2 medicina-58-01017-t002:** Multivariate-adjusted odds ratios and 95% confidence intervals for metabolic syndrome and its components across tertiles of n-3 and n-6 FA intakes by gender for Korean participants.

	Men (*n* = 1875) Tertiles of PUFA Intake			Women (*n* = 2977) Tertiles of PUFA Intake		
	1	2	3	1	2	3
**n-6 Fatty Acid**						
Range of intake(g/day)	<6.4	6.4–10.1	>10.1	<5.62	5.6–8.9	>8.9
Metabolic syndrome	1	0.81 (0.63–1.06)	0.86 (0.58–1.28)	1	0.85 (0.67–1.07)	0.89 (0.62–1.28)
High BP (SBP > 130 or DBP > 85 mmHg)	1	**0.77 (0.59–0.99)**	0.97 (0.65–1.44)	1	0.92 (0.74–1.15)	1.17 (0.83–1.66)
Increased waist circumference (>90 cm)	1	0.77 (0.59–1.01)	1.00 (0.67–1.50)	1	0.84 (0.66–1.06)	0.88 (0.61–1.26)
Increased fasting blood sugar (>100 mg/dL)	1	0.86 (0.66–1.12)	0.82 (0.55–1.22)	1	0.94 (0.75–1.17)	0.99 (0.70–1.40)
Low blood HDL cholesterol (<40 mg/dL)	1	1.03 (0.78–1.35)	1.08 (0.71–1.63)	1	**0.78 (0.63–0.96)**	**0.71 (0.52–0.97)**
High blood triglycerides (>150 mg/dL)	1	1.00 (0.77–1.31)	0.92 (0.61–1.37)	1	0.85 (0.68–1.06)	0.87 (0.61–1.23)
**n-3 Fatty Acid**						
Range of intake(g/day)	<0.87	0.87–1.40	>1.40	<0.80	0.80–1.29	>1.29
Metabolic syndrome	1	**0.70 (0.54–0.91)**	0.78 (0.54–1.14)	1	0.85 (0.68–1.06)	0.89 (0.64–1.24)
High BP (SBP > 130 or DBP > 85 mmHg)	1	**0.71 (0.55–0.92)**	0.83 (0.57–1.20)	1	1.04 (0.84–1.29)	1.24 (0.91–1.70)
Increased waist circumference (>90 cm)	1	0.87 (0.67–1.14)	0.99 (0.68–1.45)	1	**0.79 (0.63–0.99)**	0.91 (0.65–1.26)
Increased fasting blood sugar (>100 mg/dL)	1	**0.73 (0.57–0.95)**	0.70 (0.48–1.02)	1	**0.78 (0.62–0.97)**	0.86 (0.63–1.18)
Low blood HDL cholesterol (<40 mg/dL)	1	0.89 (0.68–1.17)	0.94 (0.63–1.39)	1	**0.75 (0.61–0.92)**	**0.67 (0.50–0.90)**
High blood triglycerides (>150 mg/dL)	1	0.83 (0.64–1.08)	0.76 (0.52–1.12)	1	0.82 (0.66–1.02)	0.80 (0.58–1.10)
**n-6/n-3 Ratio**						
Range of intake	<6.8	6.8–7.8	>7.8	<6.6	6.6–7.5	>7.5
Metabolic syndrome	1	0.97 (0.76–1.24)	1.12 (0.85–1.47)	1	0.79 (0.64–0.97)	1.06 (0.84–1.34)
High BP (SBP > 130 or DBP > 85 mmHg)	1	1.05 (0.83–1.34)	1.29 (0.98–1.71)	1	0.74 (0.61–0.90)	0.72 (0.58–0.91)
Increased waist circumference (>90 cm)	1	0.94 (0.73–1.21)	1.10 (0.83–1.46)	1	**0.79 (0.64–0.98)**	1.07 (0.84–1.35)
Increased fasting blood sugar (>100 mg/dL)	1	0.86 (0.67–1.09)	0.87 (0.66–1.14)	1	**0.81 (0.66–0.99)**	0.81 (0.64–1.01)
Low blood HDL cholesterol (<40 mg/dL)	1	1.02 (0.79–1.32)	0.97 (0.73–1.30)	1	0.85 (0.70–1.02)	0.92 (0.74–1.14)
High blood triglycerides (>150 mg/dL)	1	0.88 (0.69–1.13)	1.14 (0.86–1.51)	1	0.82 (0.67–1.00)	0.96 (0.76–1.20)

Adjusted for age, smoking status, alcohol use frequency, education level, employment status, physical activity, and daily intakes of energy, carbohydrates, protein, saturated fatty acids, monounsaturated fatty acids, cholesterol, and dietary fiber. Further adjusted for body mass index: high BP, increased fasting blood sugar, low blood HDL cholesterol, and high blood triglycerides. **Bold** indicates the statistical significance of the odds ratios.

**Table 3 medicina-58-01017-t003:** Multivariate-adjusted odds ratios and 95% confidence intervals for the interactions between MetS and HbA1c groups according to n-3 and n-6 fatty acid intakes and gender for Korean participants.

	Odds Ratio—HbA1c 1/3 (<5.4)			Odds Ratio—HbA1c 2/3 (5.4—5.7)			Odds Ratio—HbA1c 3/3 (>5.7)		
	Tertiles of PUFA Intake			Tertiles of PUFA Intake			Tertiles of PUFA Intake		
**Men** **(*n* = 1875)**	**1**	**2**	**3**	**1**	**2**	**3**	**1**	**2**	**3**
**n-6 Fatty Acid**									
Range of intake	<6.4	6.4–10.1	>10.1	<6.4	6.4–10.1	>10.1	<6.4	6.4–10.1	>10.1
Crude	1	**0.24 (0.17–0.34)**	**0.25 (0.18–0.35)**	1	**0.50 (0.37–0.67)**	**0.56 (0.42–0.75)**	1	1.27 (0.98–1.64)	**1.64 (1.24–2.16)**
Model 1	1	**0.57 (0.36–0.89)**	**0.57 (0.37–0.88)**	1	0.86 (0.57–1.30)	0.96 (0.64–1.42)	1	0.88 (0.62–1.26)	1.15 (0.80–1.66)
Model 2	1	**0.67 (0.42–1.09)**	**0.59 (0.36–0.96)**	1	0.98 (0.63–1.53)	1.27 (0.82–1.98)	1	0.90 (0.62–1.30)	1.16 (0.78–1.71)
Model 3	1	**0.57 (0.33–0.99)**	**0.42 (0.18–0.96)**	1	0.93 (0.57–1.55)	1.12 (0.53–2.37)	1	0.87 (0.57–1.34)	1.08 (0.54–2.16)
**n-3 Fatty Acid**									
Range of intake	<0.9	0.9–1.4	>1.4	<0.9	0.9–1.4	>1.4	<0.9	0.9–1.4	>1.4
Crude	1	**0.23 (0.16–0.32)**	**0.24 (0.17–0.35)**	1	**0.40 (0.29–0.56)**	**0.61 (0.46–0.81)**	1	1.18 (0.92–1.53)	**1.69 (1.28–2.23)**
Model 1	1	**0.50 (0.32–0.79)**	**0.54 (0.35–0.85)**	1	**0.64 (0.42–0.98)**	0.96 (0.65–1.42)	1	0.79 (0.55–1.13)	1.14 (0.79–1.64)
Model 2	1	**0.57 (0.35–0.93)**	**0.56 (0.35–0.91)**	1	0.77 (0.49–1.22)	1.24 (0.80–1.91)	1	0.78 (0.54–1.13)	1.13 (0.77–1.68)
Model 3	1	**0.47 (0.27–0.83)**	**0.37 (0.16–0.82)**	1	0.75 (0.45–1.25)	1.13 (0.55–2.34)	1	0.70 (0.46–1.06)	0.85 (0.45–1.62)
**n-6/n-3 Ratio**									
Range of intake	<6.9	6.9–7.8	>7.8	<6.9	6.9–7.8	>7.8	<6.9	6.9–7.8	>7.8
Crude	1	**0.29 (0.21–0.39)**	**0.30 (0.22–0.41)**	**1**	**0.42 (0.31–0.58)**	**0.55 (0.41–0.74)**	**1**	**1.34 (1.03–1.76)**	**1.51 (1.15–1.97)**
Model 1	1	0.87 (0.56–1.35)	0.90 (0.58–1.40)	1	**0.62 (0.41–0.94)**	0.81 (0.54–1.22)	1	0.94 (0.66–1.35)	1.06 (0.74–1.51)
Model 2	1	1.81 (0.68–1.75)	1.00 (0.62–1.60)	1	0.78 (0.50–1.21)	0.88 (0.58–1.35)	1	0.98 (0.67–1.41)	1.08 (0.75–1.55)
Model 3	1	1.22 (0.73–2.03)	1.16 (0.65–2.08)	1	0.75 (0.47–1.20)	0.79 (0.47–1.33)	1	1.17 (0.78–1.75)	**1.59 (1.00–2.52)**
**Women** **(*n* = 2977)**	**1**	**2**	**3**	**1**	**2**	**3**	**1**	**2**	**3**
**n-6 Fatty Acid**									
Range of intake	<5.6	5.6–8.9	>8.9	<5.6	5.6–8.9	>8.9	<5.6	5.6–8.9	>8.9
Crude	1	**0.13 (0.10–0.18)**	**0.10 (0.07–0.14)**	1	**0.28 (0.22–0.37)**	**0.178 (0.13–0.24)**	1	1.12 (0.89–1.41)	1.18 (0.92–1.50)
Model 1	1	**0.46 (0.31–0.69)**	**0.35 (0.23–0.53)**	1	**0.62 (0.43–0.88)**	**0.37 (0.26–0.54)**	1	0.75 (0.55–1.01)	0.79 (0.58–1.08)
Model 2	1	**0.65 (0.43–0.99)**	**0.53 (0.34–0.83)**	1	0.81 (0.55–1.18)	**0.57 (0.38–0.86)**	1	0.92 (0.67–1.28)	1.07 (0.76–1.50)
Model 3	1	0.72 (0.43–1.18)	0.65 (0.30–1.43)	1	0.95 (0.61–1.48)	0.90 (0.43–1.85)	1	0.94 (0.64–1.38)	1.02 (0.55–1.87)
**n-3 Fatty Acid**									
Range of intake	<0.8	0.8–1.3	>1.3	<0.8	0.8–1.3	>1.3	<0.8	0.8–1.3	>1.3
Crude	1	**0.15 (0.11–0.20)**	**0.10 (0.07–0.14)**	1	**0.27 (0.21–0.36)**	**0.20 (0.15–0.27)**	1	1.11 (0.87–1.40)	1.20 (0.94–1.53)
Model 1	1	**0.54 (0.37–0.80)**	**0.37 (0.24–0.56)**	1	**0.63 (0.44–0.90)**	**0.46 (0.32–0.66)**	1	0.74 (0.54–1.01)	0.80 (0.59–1.10)
Model 2	1	**0.76 (0.50–1.14)**	**0.53 (0.33–0.83)**	1	0.85 (0.58–1.24)	0.70 (0.47–1.06)	1	0.91 (0.66–1.26)	1.04 (0.75–1.46)
Model 3	1	0.83 (0.51–1.35)	0.63 (0.30–1.32)	1	1.06 (0.69–1.64)	1.17 (0.62–2.23)	1	0.88 (0.61–1.28)	0.92 (0.54–1.59)
**n-6/n-3 Ratio**									
Range of intake	<6.6	6.6–7.5	>7.5	<6.6	6.6–7.5	>7.5	<6.6	6.6–7.5	>7.5
Crude	1	**0.13 (0.09–0.18)**	**0.16 (0.12–0.21)**	1	**0.23 (0.17–0.31)**	**0.22 (0.17–0.30)**	1	0.98 (0.78–1.23)	**1.47 (1.16–1.86)**
Model 1	1	**0.60 (0.40–0.90)**	0.71 (0.48–1.04)	1	**0.51 (0.35–0.73)**	**0.49 (0.340–0.70)**	1	0.67 (0.49–0.91)	1.01 (0.74–1.39)
Model 2	1	0.84 (0.54–1.30)	1.07 (0.71–1.63)	1	**0.65 (0.44–0.95)**	**0.61 (0.42–0.90)**	1	0.79 (0.57–1.08)	1.24 (0.89–1.73)
Model 3	1	0.88 (0.56–1.39)	1.20 (0.73–1.97)	1	0.67 (0.44–1.00)	0.68 (0.44–1.07)	1	0.83 (0.60–1.15)	**1.52 (1.03–2.23)**

Adjusted for (Model 1): age; (Model 2): Model 1 + smoking status, alcohol use frequency, education level, employment status, lowest income quartile, and physical activity; (Model 3): Model 2 + daily intakes of energy, carbohydrates, protein, saturated fatty acids, monounsaturated fatty acids, cholesterol, and dietary fiber. **Bold** indicates the statistical significance of the odds ratios.

## Data Availability

All data in the present study are available from the corresponding author upon reasonable request.

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
