# Peer review of "Relation of Dietary n-3 and n-6 Fatty Acid Intakes to Metabolic Syndrome in Middle-Aged People Depending on the Level of HbA1c: A Review of National Health and Nutrition Survey Data from 2014 to 2016"

_medicina, 2022, doi:10.3390/medicina58081017_

Round 1

Reviewer 1 Report

In the manuscript »Relation of dietary n-3 and n-6 fatty acids intake to metabolic syndrome depending on the level of HbA1c: review of national health and nutrition survey data from 2014 to 2016«, the authors present a study investigating the influence of omega 3 and omega 6 fatty acids intake on metabolic syndrome parameters. I am of opinion, that in the present form, it is not good enough to be published and that substantial revision is needed before reconsidering it for a potential publication.

My comments:

11. Introduction section is too long and should be focused on the parameters investigated in the study.

12. In general, the article is too long and not clear enough, and therefore difficult to read. It should be shortened for at least of one third.

33. Methodology is based on dietary intake of some nutrients and should be decribed in detail.

44.  The size of individual groups presented in tables 1 to 3 should be presented.

55. The novelties of the study should be exposed. Namely, the metabolic syndrome parameters in the relation to individual nutrients have been already extensively investigated (e.g. Jang H, Park K. Clin Nutr 2020).

66.  In Abstract, the prevalence of metabolic syndrome has to be cited in Results section.

77. The definition of metabolic syndrome according to National Cholesterol Education Program has to be checked (e.g. waist circumference cut-off values).

88. The HbA1c, representing the basics of the study, is not an equivalent to insuline resistance.

99.  Insulin sensitivity is not a hormone. In addition, angina is a symptom, not a disease.

110.   Biases in complex diseases like metabolic syndrome have to be discussed.

111.   Comment on data in table 1 where all important nutritional parameters in women with metabolic syndrome are lower than in nonmetabolic ones, and the difference is statistically significant.

Author Response

Thank you for taking the valuable time to review. We have tried to correct what you have pointed out to suit your intentions. Please checking the ms word file. We appreciate the opportunity to improve our article. 

Reviewer 2 Report

I read this study by Park et al. with great interest. The study aimed to evaluate Relation of dietary n-3 and n-6 fatty acids intake to metabolic syndrome depending on the level of HbA1c: review of national health and nutrition survey data from 2014 to 2016. The result of this study postulated that consistent and regular dietary intake of n-3 and n-6 fatty acids may contribute greatly to prevent or to treat metabolic syndrome. Despite of its great contribution to expand our current knowledge, this idea suffers from several weaknesses to be used to make a valid and trustworthy conclusion.

- Check keywords in the MeSH.

- Introduction is too long and boring. Please try to keep it concise and well-developed. Also, the introduction section should be continuous as well as highlighting the goal and novelty of the study.

- please discuss your novelty.

- Please describe the possible reasons for choosing 40-64 years old years of age as your inclusion criteria.

- This study is retrospective or perspective?

- Details on the method of recruitment are completely missed. Please describe it.

- What are the data extract’s center characteristics? Is it governmental or private, is it referral or not referral, and so on, discuss more it.

- Please categorize the sample size according to the energy intake (500-6000 kcal/day). It’s a wide range.

-Why the first measurements of blood parameters were ignored?

- Variables for Life Styles and Demography section is not applicable and it’s better to summarize it.

- How physical activities were evaluated?

- Did you check normal assumption? Please mention it.

- Please address the reasons why you did not evaluate dietary supplement such as vitamin E, vitamin d, and antioxidants to exclude subjects or as exclusion criteria

- The overall novelty of the study is dubious in terms of making a significant advance in our knowledge. It’s a big problem.

Author Response

(The authors gave the same response as above.)

Reviewer 3 Report

Relation of dietary n-3 and n-6 fatty acids intake to metabolic syndrome depending on the level of HbA1c: review of national health and nutrition survey data from 2014 to 2016

Dear Authors,

Congratulations on conducting this interesting study. This manuscript tries to reveal the relationship between the HbA1c, metabolic syndrome and consumption of n-3/n-6 fatty acids. The consumption of PUFA and their association is well known for health benefits and the author successfully has reached the conclusion of their study. 

The following are my comments and suggestions:

Abstract and Introduction:

  1. My most of comments are on the consistency of data presentation. The manuscript needs to be edited for punctuation and data presentation. Kindly look at the file for your reference.
  2. Abstract. This is too wordy and unhelpful to conclude changes among parameters. Please mention changes in the percentage form to show the degree of changes that appeared in the data.
  3. The introduction section needs to rewrite heavily so that reader can illustrate the information in the introduction what as the problem, history, and Author's hypothesis. Three paragraphs i.e., fatty acids, metabolic syndrome and insulin resistance can emerge in one and their information and conclusion can be provided briefly in one paragraph.   

Methods:
1. What was the status of the liver health of candidates in this study? Did liver enzymes examine? Results from urine test for this study? Kindly provide liver enzyme-related data.
2. Why did 40- and 64-year-old patients selected for this study in method sections? Aged and young adults could be taken part in the study as well?
3. Statistical analysis – n3 and n6 intake against daily intake of calories, carbohydrates, protein, SFA, MUFA, cholesterols, and dietary fibre – did the author find any correlation among them? Interesting observation?

Results:
1. All tables are presented very well. Consistency, punctuation, and font size as well. However, the p-value should be written in italic format.
2. More preciously, few and the most important results can be presented in graphical form. The significant p-values should be highlighted in the graphs so that the reader can understand data analysis outcomes easily and proficiently. Instead of using the numbers in the tables, please put a few graphs and diagrams to make the manuscript more enjoyable to read. 

Discussion:
So, based on these conclusions, how do we proceed with managing metabolic syndrome, especially, in the liver? What dose of PUFA should be taken to maintain health?

Can the same observation be seen non-Korean populations?

Will this observation be considerable for both type 1 and type 2 diabetes as well?

The author stated a few claims. Please give references for your statements. Please elaborate and discuss. 

Please check out the reference list. The Journal name is not consistent throughout the list. Kindly correct it. 

Author Response

(The authors gave the same response as above.)

Round 2

Reviewer 2 Report

Dear Editor

The authors have done their best to address all of my comments and responded accordingly and appropriately. Thus, the manuscript is greatly improved from the original version. 

Author Response

Thank you again, for taking your valuable time and effort to improve our article.

Reviewer 3 Report

Dear Author, 

Thank you for explaining and answering my questions. All comments and corrections have been clarified and are understood. However, there are a few grammatical errors in the document that must be corrected as efficient as possible. Please see the attached file for my suggestions.

Author Response

Dear Reviewer,

Thank you for taking the valuable time to review again. 

We would appreciate it if you could refer to the file we attached.

We look forward to hearing from you. 

Yours sincerely, 

Sung-Kyung Kim, MD, PhD
On behalf of the co-authors.
